# Breaking Barriers: Artificial Intelligence Interpreting the Interplay between Mental Illness and Pain as Defined by the International Association for the Study of Pain

**DOI:** 10.3390/biomedicines11072042

**Published:** 2023-07-20

**Authors:** Franciele Parolini, Márcio Goethel, Klaus Becker, Cristofthe Fernandes, Ricardo J. Fernandes, Ulysses F. Ervilha, Rubim Santos, João Paulo Vilas-Boas

**Affiliations:** 1Center for Rehabilitation Research (CIR), School of Health, Polytechnic Institute of Porto, Rua Dr. António Bernardino de Almeida, 400, 4200-072 Porto, Portugal; rss@ess.ipp.pt; 2Porto Biomechanics Laboratory, University of Porto, 4200-450 Porto, Portugal; gbiomech@fade.up.pt (M.G.); ricfer@fade.up.pt (R.J.F.); ulyervil@usp.br (U.F.E.); jpvb@fade.up.pt (J.P.V.-B.); 3Center of Research, Education, Innovation and Intervention in Sport, Faculty of Sport, University of Porto, 4200-450 Porto, Portugal; 4Faculty of Psychology and Educational Sciences of the University of Porto, 4099-002 Porto, Portugal; up201802555@up.pt; 5Laboratory of Physical Activity Sciences, School of Arts, Sciences and Humanities, University of São Paulo, São Paulo 03828-000, Brazil

**Keywords:** lower back pain, mental illness, pain catastrophizing, artificial intelligence (AI), motor disability, pain perception, affective components, central nervous system, emotions, peripheral nervous system, sensation

## Abstract

Low back pain is one of the main causes of motor disabilities and psychological stress, with the painful process encompassing sensory and affective components. Noxious stimuli originate on the periphery; however, the stimuli are recombined in the brain and therefore processed differently due to the emotional environment. To better understand this process, our objective was to develop a mathematical representation of the International Association for the Study of Pain (IASP) model of pain, covering the multidimensional representation of this phenomenon. Data from the Oswestry disability index; the short form of the depression, anxiety, and stress scale; and pain catastrophizing daily questionnaires were collected through online completion, available from 8 June 2022, to 8 April 2023 (1021 cases). Using the information collected, an artificial neural network structure was trained (based on anomaly detection methods) to identify the patterns that emerge from the relationship between the variables. The developed model proved to be robust and able to show the patterns and the relationship between the variables, and it allowed for differentiating the groups with altered patterns in the context of low back pain. The distinct groups all behave according to the main finding that psychological and pain events are directly associated. We conclude that our proposal is effective as it is able to test and confirm the definition of the IASP for the study of pain. Here we show that the fiscal and mental dimensions of pain are directly associated, meaning that mental illness can be an enhancer of pain episodes and functionality.

## 1. Introduction

In several countries, low back pain is referred to as the major source of musculoskeletal complaints, with a high impact on health and the economy due to the limitations and disability it imposes on individuals and their need for healthcare and absenteeism from work [1]. The process that leads to the perception of pain is a complex succession of peripheral and central neural system activities, including modulation at different levels. Despite the essentially peripheral origin of low back pain, its cause is not identified in 85% of the cases [1,2]. The pain process encompasses sensory, cognitive, and affective components [3], with this last component including feelings of annoyance, sadness, anxiety, and depression in response to a harmful stimulus [4]. Brain activity in patients with low back pain for two months showed activation in the insular cortex, thalamus, anterior cingulate cortex, and prefrontal cortex.

For over a decade, it has been noted that those who suffer from long-term lower back pain exhibit activity in certain areas of the brain, specifically the perigenual anterior cingulate and medial prefrontal cortexes, as well as the amygdala [3,4]. This implies that as acute pain becomes chronic, there may be a shift towards emotional pathways instead of just sensory ones. Furthermore, the experience of pain, as well as anxiety and depression, is often intertwined with the idea of suffering [5]. The association of pain with the subjects’ psychological state has been investigated [6,7,8], and there seems to be a consensus that psychological and social factors are fused in biopsychosocial processes that characterize chronic pain [9].

In recent years, there has been a growing recognition of the close relationship between pain and mental illness [10,11]. It has become increasingly clear that mental health conditions, such as depression, anxiety, and stress, can significantly influence the experience and perception of pain [10]. Conversely, chronic pain can also have a profound impact on mental well-being, leading to increased levels of psychological distress and impairment [6]. Recent studies elucidate those different types of chronic pain conditions such as fibromyalgia and low back pain and chronic pain conditions from underlying medical conditions such as post-trauma, neuropathic, and musculoskeletal pain have distinct pathogenic pathways [12,13]. So perhaps chronic back pain is best understood in the framework of pain perception, including cognitive, emotional, and social components; therefore, the association of mental health and pain perception appears to be a logical association [13,14]. Understanding and addressing this intricate relationship between mental illness and pain is crucial for providing comprehensive and effective care to individuals experiencing pain.

As of 2020, the meaning of pain has been redefined by the International Association for the Study of Pain (IASP). According to the new definition, pain is an unpleasant sensory and emotional experience associated with or similar to that associated with actual or potential tissue damage [10]. Those who have extensive knowledge in pain-related fields, including clinical and fundamental science, decided on the model by examining existing definitions and annotations and deciding whether they still apply or need modification. Although it seems to be very well accepted in the community, a global multivariate model can provide more robust support for what is currently the most accepted definition. If such a model considers, with the respective weights, the interaction of a set of variables involved, this multivariate phenomenon will certainly be better understood, and consequently, more accurate and adequate diagnostic and therapeutic tools will be developed.

To gain a deeper understanding of the complex interplay between mental illness and low back pain, researchers have turned to mathematical modeling and artificial intelligence as powerful tools [15,16,17,18]. Machine learning and deep learning algorithms offer the ability to analyze small and large numbers of data and discover hidden patterns and associations that may not be evident through traditional statistical approaches [19,20]. One of the main advantages of using artificial intelligence to study the relationship between mental ill-health and low back pain is its ability to capture and analyze multiple dimensions of pain [21]. Traditional research methods often focus only on the physiological aspects of pain, such as measuring pain intensity or identifying biomarkers. While these aspects are undeniably important, they provide only a partial understanding of the pain experience [18,22].

Mathematical modeling can provide a more comprehensive representation of pain by integrating functional, psychological, and emotional factors into the analysis, and artificial intelligence algorithms allow researchers to analyze complex and heterogeneous data and can help identify patterns and relationships between variables, determining the relationship between low back pain and its interaction with mental illness [23,24,25,26]. Pain is a subjective experience, the evaluation of which depends largely on self-reported measures. These measures often include questionnaires, surveys, and diaries to capture people’s perceptions, emotions, and behaviors related to pain. AI algorithms can process and analyze these data sources, generating meaningful insights and identifying patterns that help understand how pain and mental illness relate to and affect an individual’s quality of life [27].

In order to improve understanding of the link between low back pain and psychological conditions, and to aid in better assessment and decision-making by healthcare professionals, artificial intelligence has shown great efficacy [20]. Artificial intelligence algorithms can analyze behavioral, language, and emotional functional patterns captured in digital data, such as text messages, social media, or electronic health records, and identify indicators of emotional pain and distress [28]. This information can be used to develop low back pain tracking tools and continuous monitoring for more timely and individualized interventions.

Therefore, the aim of the current study was two-fold: (i) to develop a mathematical representation based on a multivariate model to elucidate the relation between low back pain and biopsychosocial aspects and (ii) to identify subpopulations that present deviations from the pattern that emerged. We hypothesize that it is possible to test the IASP concept of pain through a mathematical representation (evidencing its coherence) and that there is a strong relationship between mental health and the way the subject copes with the experience of pain and its functional consequences.

## 2. Materials and Methods

Details of the study design are presented, including methodological approaches that we use to analyze the complex interactions of low back pain phenomena and try to understand the underlying patterns and relationships, with the auxilium of mathematical modeling and the algorithms of artificial intelligence [28,29,30]. The methodological design of our study allows us to provide a comprehensive view of the research process and aims to ensure the validity and reliability of the results obtained. With targeted methods, we intend to expand our knowledge in this field, advancing our understanding of the interaction between low back pain and mental illness [20].

This was a cross-sectional observational study approved by the ethics committee of the School of Health of the Polytechnique of Porto (CE0092B), and the study objectives and procedures were developed and conducted in accordance with the guidelines of the Declaration of Helsinki. Volunteers consented to participate in the study through their informed consent form. The sample consisted of 1.021 young adults (73% females), aged between 18 and 35 years (24.68 ± 1.5 years, height 167.9 ± 0.1 m, and weight 65.8 ± 3.5 kg). The exclusion criteria were <18 years old, >35 years old, or not completing the survey. The research involved the Center for Rehabilitation and Research (CIR) of the Higher School of Health of the Polytechnic of Porto and the Laboratory of Biomechanics of the University of Porto (LABIOMEP).

### 2.1. Data Collection

The survey focusing on the relation of low back pain with psychological variables in young adults was created with Lime Survey version 3.28.56 + 230404, an online survey application software written in pre-processed Python text. Data were collected through online auto-completion on the Lime platform in the period from 8 June 2022, to 8 April 2023. The link to access the survey was disseminated through the institutional emails of the Polytechnic of Porto and the University of Porto to the entire academic population and also in social networks. Participants provided information related to gender, mass, age, height, sociodemographic information, the existence of medical diagnosis of psychiatric disorder, and the frequency of episodes of low back pain in six weeks.

### 2.2. Instruments

The Oswestry disability index I [31] was used in the survey as a specific instrument that measures the impact of back pain on daily living activities (particularly regarding pain intensity, lifting weights, social interaction, sitting, standing, traveling, sex life, sleeping, walking, and caring). It is composed of 10 questions with 6 alternatives (each ranging in scores from 0 to 5). The first question assesses the intensity of pain, while the others score the pain impact on daily activities (such as personal care, lifting weights, walking, sitting, standing, sleeping, social activities, and mobility). The total score is multiplied by the number of questions answered and then multiplied by 5, with the result expressed as a percentage ([score ÷ (number of questions answered × 5)] × 100). The scores are classified as minimal, moderate, and severe disabilities (0–20, 21–40, and 41–60%, respectively); disabled (61–80%); and bedridden (81–100%).

The short form of the depression, anxiety, and stress scale [32] was also used (including 21 items) and was designed to assess depression, anxiety, and stress domains (each one being represented by 7 items). Participants rated each item on a 0 (“did not apply to me at all”) to 3 (“applied to me very much or most of the time”) scale. Each domain is represented by a subscale score (the sum of the item responses for that subscale multiplied by two to be comparable with the original 42-item depression, anxiety, and stress scale). This instrument was previously validated and considered reliable [32], with a high score representing worse depression, anxiety, or stress. Cut points for normal, mild, moderate, severe, and extremely severe score classification, based on population norms, are provided in its manual. Classification symptoms are rated as 0–10 (normal), 11–18 (mild), 19–26 (moderate), 27–34 (severe), and 35–42 (extremely severe) for stress; 0–6 (normal), 7–9 (mild), 10- 14 (moderate), 15–19 (severe), and 20–42 (extremely severe) for anxiety; and 0–9 (normal), 10–12 (mild), 13–20 (moderate), 21–17 (severe), and 28–42 (extremely) severe for depression.

Pain catastrophizing daily [33] is a questionnaire with 14 points that aims to assess disasters in the last 24 h, whose items were also rated by our participants on a scale of 0 (“never”) to 4 (“always”). The total score was calculated as the sum of the item responses (range 0–56), with higher scores representing greater catastrophizing of pain. The use of the daily catastrophe questionnaire may lead to greater analytical accuracy in research, health tools and platforms, and studies of psychosocial diaries that seek to understand the adaptive mechanisms of pain.

### 2.3. Anomaly Detection Structure

Anomaly detection refers to the problem of finding data patterns that do not conform to the expected behavior [23]. In the current research, a dataset of 1.021 volunteers was used to model the relationship patterns between the low back pain-related variables. An artificial neural network structure with two hidden layers was trained, with each of the hidden layers including tangent hyperbolic transfer and a logistic sigmoid with 20 neurons, and fully connected (Figure 1). The input layer was composed of socioanthropometric dimension-related variables (age, sex, body mass, height, and body mass index) and data from the Oswestry disability index I [31]; depression, anxiety, and stress scale [32]; and pain catastrophizing daily [33] questionnaire scores. The output layer contained the same information but with a randomized subject order. The output space consisted of a “1” or “2” binary classification, indicating “no change” and “change” in the general functional profile (respectively). The learning algorithm used was Bayesian regularization. The dataset was randomly split into 80% of samples for training and 20% for testing.

After 727 epochs, a mean square error performance value of 0.001 was obtained. The accuracy achieved after training equals R = 0.9903, 0.9625, and 0.9846 in training, in the test, and for all samples (respectively). Then, data of all subjects were simulated using the model obtained, and the estimates were compared with the real data through a single linear regression, where the target was the dependent variable and the output was the independent variable.

Subsequently, three subgroups were created, determined by the position of the R in relation to the 25th and 75th percentiles (the first formed by subjects with values < 25th percentile; the second, from 25th to 75th percentiles; and the third, >75th percentile). Since data did not show a normal distribution, the between-group comparison was performing using the Kruskal–Wallis test (with the pairwise comparison conducted using the Mann–Whitney U test adjusted with the Bonferroni correction).

## 3. Results

The model seems to capture some interesting differences between the groups (Figure 2), showing a relationship between the variables of number of lower back pain events in a 6-week period (*p* = 0.001), medical diagnosis of lower back pathology (*p* = 0.002), ODI (*p* = 0.001), age (*p* = 0.030), and anthropometric data and correlated with the psychological variables, stress (*p* = 0.001), anxiety (*p* = 0.001), depression (*p* = 0.001) and catastrophizing in the last 24 h in episodes of low back pain (*p* = 0.001). The results are expressed as the multiplication factor (MF) of each condition that is multiplied by the constant value (as mean) of each variable.

### Statistical Analysis

Table 1 shows the difference between groups and effect size regarding each variable, followed by median and interquartile range values. The GPower 3.1.7 software (University of Kiel, Kiel, Germany) was used to calculate the effect size (ES) and determine the power of analysis using the Mann–Whitney U, followed by Cohen’s d criterion (small: >0.2; moderate: >0.50; large: >0.80) [34]. No differences were found between groups regarding body mass, height, gender, or mental illness diagnosis. Lumbar pathology was higher in group 1 than in group 3 (*p* < 0.001) and in group 2 than in group 3 (*p* = 0.039), and low back pain events presented a similar behavior, i.e., group 1 > 2 (*p* = 0.001) and group 1 > 3 (*p* = 0.003). The psychological variables differed between groups, with stress being higher in group 1 than in group 2 and in group 1 than in group 3 (both for a *p* < 0.001); anxiety being higher in group 1 than in group 3 (*p* < 0.001), in group 1 than in group 2 (*p* < 0.001), and in group 2 than in group 3 (*p* = 0.031); depression displaying higher values in group 2 than in group 3 (*p* = 0.019), in group 1 than in group 3 (*p* < 0.001), and in group 1 than in group 2 (*p* < 0.001); and pain catastrophizing daily showing the results of group 1 > 2 and group 1 > 3 (both for a *p* < 0.001) due to its epistemological proximity. Given that psychological variables are factors that can exacerbate pain, the higher Oswestry disability index I values in group 1 than in 3 (*p* = 0.050), showing a mild difficulty in lumbar functionality, are not surprising.

These findings provide valuable information about the factors that contribute to low back pain in young adults and emphasize the importance of considering physiological and psychological aspects in understanding and managing this condition.

## 4. Discussion

Pain and mental illness together should be part of an integrated treatment approach. It should involve a multi-professional team, with a combination of physical interventions, such as exercise, physical therapy, medication to manage pain, and psychological interventions, to address the mental status and improve the functional status [22]. Therefore, research in this area, with the aid of multivariate models, is of great importance, as it allows the identification of risk and protection factors associated with pain and mental illness. These include genetic, environmental, psychosocial, and behavioral factors that may influence the development of these conditions. Understanding these factors enables the implementation of more effective preventive strategies and the development of targeted interventions, playing an important role in reducing the stigma associated with these conditions [35,36,37].

The mathematical modeling we used in our study can lead to advances in the delivery of care from all areas of healthcare. Using effective screening artificial intelligence algorithms, unusual patterns in the frequency, intensity, or duration of low back pain over time can be identified, which is useful for identifying episodes of severe acute pain or significant changes in pain patterns. This can be applied to identify specific activities, postures, or movements that lead to a significant increase in pain [29,38]. This information can help identify behaviors or situations that should be avoided or changed to improve pain management, and thus identify triggers associated with low back pain episodes and their physical and mental functions.

We found evidence of a relationship between the repetition of traumatic events and physical and mental functioning, particularly stress, anxiety, and ultimately depression. According to the literature and the data obtained in this study, the repeated experience of pain can have a significant impact on a person’s daily functioning and can also increase the risk of developing or worsening depressive symptoms [30,39]. Recurrent or persistent pain can limit a person’s ability to carry out daily activities, such as work, exercise, socializing, and self-care. In the case of persistent pain, it can affect sleep, energy, mood, and quality of life, leading to symptoms of depression [20,31]. Mental and emotional health plays a significant role in the experience and perception of pain, and addressing these aspects can bring substantial benefits to patients [26,38]; thus, this study has significant potential by exploring the direct relationship between musculoskeletal pain and mental ill-health.

Considering that pain is an unpleasant sensory and emotional experience associated with (or resembling) actual/potential tissue damage, there should be quantifiable emotional variables that allow transcribing it into a mathematical model. Moreover, due to the sensory–motor nature of this phenomenon, movement measures or scores should be included in the model. Data from human movement biomechanical variables are commonly heterogeneous and form a large volume of information, making it difficult to treat them using inferential statistics. However, advanced analytical techniques used to evaluate informative data features and model underlying relationships that cannot be treated with traditional statistics can increase the research quality [29,40]. For a more global understanding of low back pain multivariate phenomena, widely used artificial intelligence tools [41] should be employed. Aiming to mathematically represent the IASP [10] definition of pain using an artificial neural network approach, based on the current study results, we advocate that it is possible to mathematically model and represent it.

The mathematical model that we have presented processed information from 1.021 volunteers allowing us to assess the linear and nonlinear relationships between variables that construct the phenomenon. It showed a very robust final performance and identified the subpopulations that presented deviations from the pattern in the context of low back pain and biopsychosocial aspects [29,41]. The relationships between the variables that emerged from this model can be seen in the group profiles. An interesting fact in the group < 25th percentile is that the lumbar pathology diagnosis is closely linked to the depression, anxiety, and stress scale-related variables [32]; pain catastrophizing daily [33]; and low back pain events, promoting a slight functional incapacity of the individual. It seems that this functional incapacity makes it difficult for individuals to carry out their usual activities [24], eventually leading to social isolation and having a major negative effect on individual well-being.

The current study results show an interdependence of variables, meaning that, for example, our oldest group also has a higher prevalence of diagnosis of lumbar problems and low back pain flares, as well as scoring worse on depression, anxiety, and stress scale and pain catastrophizing daily and Oswestry disability index I surveys. However, our data cannot give a good explanation about the underlying mechanism, i.e., if the low back pain flares lead to worse psychological variables or if the psychological impairment leads to perception and aggravation of the pain (leading to seeking medical diagnosis).

The relationship between low back pain, psychological distress, and mild functional disability observed by us is in line with previous data that identified high levels of pain intensity associated with poor psychological and physiological capacity and high levels of anxiety and depression [42]. Based on the current study results and on the literature, it is possible that the mental disorder in low back pain may be a predictor of reduced functionality [32,43] and to hypothesize that individuals with a medical diagnosis of lower back pathology have a higher number of lower back pain episodes over a six-week period and higher levels of pain catastrophizing.

Our data are in line with a study with 84 patients with rotator cuff tears that were evaluated for the presence of differences in pain, function, and/or psychological distress associated with pain and analyzed for the association between psychological distress with shoulder pain and function during adjustment for cuff tear severity [43]. Results demonstrated that baseline psychological distress is related to patients’ pain and shoulder function more than the diagnosis of rotator cuff tears, suggesting that the size and severity of the lesion are not fully related to symptoms (e.g., pain and functional limitation) but rather to psychological distress [43,44]. Anxiety and avoidance can cause an inflated sense of pain [45,46], while fear of pain influences short-term pleasure seeking [25] due to pain’s catastrophic aftermath [47,48]. These behavioral patterns are not connected to the disease at hand.

Based on these statements, a study in mice examines whether long-term associations with remembering fear stored in neural engrams in the prefrontal cortex can determine how painful episodes evolve into later-life painful experiences [49]. It was evidenced that long-term fear memory is associated with pathological changes in nociceptive sensitivity following tissue injury, a key feature of pathological pain disorders and known to be regulated by the cortex [50]. Pain and fear are independent behavioral states that are interrelated [46,51], with fear acutely potentiating the perception of pain [49] that is fundamental to survival. It was concluded that a painful experience could encode a memory of fear (that will be stored in a discrete and specific cohort of prefrontal cortical neurons). This will be subjected to reactivation after exposure to a new painful stimulus in future life events, and as a result, it will produce an intensification of pain perception [50,52].

According to the approach mentioned above and the data from our study, it can be underlined that the catastrophizing of pain leads to excessive fear of pain, and the associative long-term memory of fear induced by previous exposure to pain may also be a critical predisposing factor for pain chronicity [51,52]. Thus, the fear of pain can provoke avoidance of motion behaviors and exacerbate pain in the long term, implying an increase in the functional disability of the individual. It is important to address that the relationship between pain, functionality, and depression is bidirectional.

This study has some limitations. Data from self-completion questionnaires rely on the accuracy and honesty of participants’ responses. However, these responses may be subject to self-report bias, where participants may provide inaccurate or biased responses. This may occur due to memory problems, lack of understanding of the questions, or desire to please the researcher or hide certain information, besides not having a face-to-face and objective verification of the data provided by the participants. We took these limitations into consideration when constructing and applying the survey and interpreting the study results. We understood the possible sources of bias, which helped us to assess the validity and reliability of the results obtained. In addition, we combined different methods of complementary analysis which allowed us to strengthen the conclusion of our study.

## 5. Conclusions

In view of the above, we conclude that it is possible to validate and confirm the IASP definition of pain through mathematical modeling. The identified subpopulations showed a direct relationship between pain and mental illness, with these two inducing greater disabilities. Even if these results may help to improve the understanding of mental illness as a possible enhancer of pain episodes and functionality, future studies evaluating other variables, like the level of physical activity and the sedentary behavior of the subjects, are required to better understand the mentioned association.

## Figures and Tables

**Figure 1 biomedicines-11-02042-f001:**
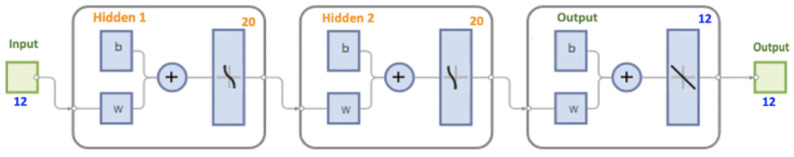
Used artificial neural network structure.

**Figure 2 biomedicines-11-02042-f002:**
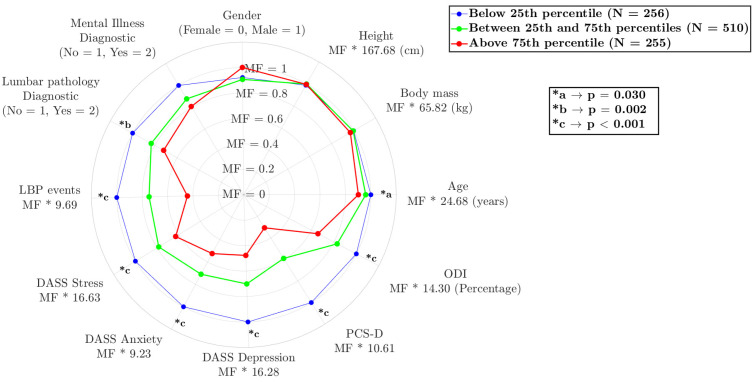
Comparison of different variables among the three subgroups, with the variable names being followed by the value to be multiplied by the multiplication factor. Legend: stress (DASS Stress), anxiety (DASS Anxiety), depression (DASS Depression) scale short; pain catastrophizing daily (PCS-D); and Oswestry disability index (ODI).

**Table 1 biomedicines-11-02042-t001:** Comparison of different variables among the three subgroups. Legend: Cohen’s d test value; stress (DASS Stress), anxiety (DASS Anxiety), depression (DASS Depression) scale short; pain catastrophizing daily (PCS-D); Oswestry disability index (ODI); group 1 (G1); group 2 (G2); group 3 (G3), * has binary values described in the results session, *p*-value, and Cohen’s d test value.

Variables	Comparison		*p*	*d*
Age	G1–22 (8)	G2	0.322	1.494
		G3	0.000	0.212
	G2–22 (8)	G1	0.322	1.494
		G3	0.008	0.179
	G3–21 (5)	G1 G2	0.000	0.212
			0.008	0.179
Body mass	G1–62 (21)	G2	0.012	1.503
		G3	0.001	0.301
	G2–63 (16)	G1	0.012	1.503
		G3	0.437	0.191
	G3–60 (12)	G1	0.001	0.301
		G2	0.437	0.191
Height	G1–167 (11)	G2	0.801	1.403
		G3	1.000	0.177
	G2–165 (12)	G1	0.801	1.403
		G3	1.000	0.054
	G3–165 (12)	G1	1.000	0.177
		G2	1.000	0.054
Gender	G1 *	G2	1.000	0.041
		G3	1.000	0.138
	G2 *	G1	1.000	0.041
		G3	1.000	0.034
	G3 *	G1	1.000	0.138
		G2	1.000	0.034
Mental illness diagnosis	G1 *	G2	0.054	0.111
		G3	0.060	0.158
	G2 *	G1	0.054	0.111
		G3	1.000	0.033
	G3 *	G1	0.060	0.158
		G2	1.000	0.033
Lumbar pathology diagnosis	G1 *	G2	0.041	1.421
		G3	0.000	0.200
	G2 *	G1	0.041	1.421
		G3	0.142	0.136
	G3 *	G1	0.000	0.200
		G2	0.142	0.136
LBP events	G1–4 (9)	G2	0.000	1.469
		G3	0.000	0.260
	G2–4 (4)	G1	0.000	1.469
		G3	0.028	0.280
	G3–4 (4)	G1	0.000	0.260
		G2	0.028	0.280
DASS—stress	G1–14 (16)	G2	0.000	1.686
		G3	0.000	0.517
	G2–10 (10)	G1	0.000	1.686
		G3	0.015	0.424
	G3–10 (8)	G1	0.000	0.517
		G2	0.015	0.424
DASS—anxiety	G1–8 (12)	G2	0.000	1.677
		G3	0.000	0.506
	G2–4 (6)	G1	0.000	1.677
		G3	0.021	0.413
	G3–2 (8)	G1	0.000	0.506
		G2	0.021	0.413
DASS—depression	G1–12 (20)	G2	0.000	1.757
		G3	0.000	0.596
	G2–6 (10)	G1	0.000	1.757
		G3	0.001	0.510
	G3–6 (8)	G1	0.000	0.596
		G2	0.001	0.510
PCS-D	G1–2 (15)	G2	0.000	1.523
		G3	0.000	0.326
	G2–1 (8)	G1	0.000	1.523
		G3	0.000	0.218
	G3–1 (5)	G1 G2	0.000 0.000	0.326 0.218
ODI	G1–10 (18)	G2 G3	0.015	1.416
			0.000	0.193
	G2–10 (16)	G1	0.015	1.416
		G3	0.036	0.072
	G3–8 (10)	G1 G2	0.000 0.036	0.193 0.072

## Data Availability

The data presented in this study are available on request from the corresponding author. The data are not publicly available due to privacy.

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
