# Peer review of "Breaking Barriers: Artificial Intelligence Interpreting the Interplay between Mental Illness and Pain as Defined by the International Association for the Study of Pain"

_biomedicines, 2023, doi:10.3390/biomedicines11072042_

Round 1
Reviewer 1 Report
Referee’s Report on
“Mathematical representation of the IASP definition of pain – a model based on physical and psychological quantitative variables”
by
Franciele Carvalho Santos Parolini, Márcio Fagundes Goethel, Klaus Magno Becker, Cristofthe Fernandes, Ricardo Jorge Fernandes, Ulysses Fernandes Ervilha, Rubim Santos, and João Paulo Vilas-Boas
Summary
In this paper, the authors developed a mathematical representation of the International Association for the Study of Pain (IASP) model of pain, covering the multidimensional representation of the phenomenon.
General comments
The paper is well-written and deals with an interesting topic. Some minor comments follow:
- Please do not use abbreviations in the title.
- In lines 143-145, the authors stated that data from all subjects were simulated using the model obtained, and the estimates were compared with the real data through a single linear regression. More details are needed in the single linear regression (e.g., which are the dependent and independent variables used).
- A paragraph with the limitations of the study is missing.
My overall opinion is that the paper may be published after corrections have been made.
Minor editing of English language required
Author Response
Thank you for your considerations. The answers are in the attached file.
Please see the attachment.

Reviewer 2 Report
Validation and confirmation of the definition of pain by IASP through mathematical modelling is an interesting topic. The authors used an artificial neural network structure to identify the patterns that emerge from the relationship between the variables.
In the present paper there are several points that remain unclear:
Why did they use a neural network approach? There is no high dimensional data, and there is not a large amount of training data. In my understanding, there is tabular data, where deep learning is rarely the correct model choice.
The division in training and test data is unclear.
Why only young adults as participants? These are not the majority in low back pain.
What does “Lumbar pathology” mean? This should be specified. As such, lumbar pathology is not probable in young persons.
Line 198: “An interesting fact in the group < 25th percentile is that the lumbar pathology diagnosis is closely linked to the depression anxiety stress scales related variables [12], pain catastrophizing daily [13] and low back pain events, promoting a slight functional incapacity of the individual.” This is a trivial conclusion. It applies for many diseases, that the presence of pathology is related to depression.
There are missing words, e.g.
Line 74 XXXX?
Line 80 XXXX?
Line 87 XXXX?
Author Response

(The authors gave the same response as above.)
